# Antecedents of Workplace Psychological Safety in Public Safety and Frontline Healthcare: A Scoping Review

**DOI:** 10.3390/ijerph22060820

**Published:** 2025-05-23

**Authors:** Emily Ip, Rani Srivastava, Liana Lentz, Sandra Jasinoski, Gregory S. Anderson

**Affiliations:** Thompson Rivers University, Kamloops, BC V2E 0C8, Canada

**Keywords:** psychological safety, public safety, healthcare, leadership, hierarchy, workplace climate

## Abstract

Workplace psychological safety (PS) has been recognized as crucial in mitigating organizational stressors and enhancing positive workplace outcomes, particularly in high-risk occupations such as healthcare and public safety. This scoping review aims to synthesize the existing literature on psychological safety in high-risk workplaces to identify key antecedents, workplace enhancements, and research gaps. A systematic search of studies published between 2010 and January 2025 was conducted across multiple databases, including CINAHL, Medline, ERIC, JSTOR, PsycInfo, Business Source Ultimate, and Sociological Abstract. Inclusion criteria focused on adult workers in high-risk workplace environments. Following screening and eligibility assessments, 16 studies from six countries were selected for analysis. Data were extracted and thematically analyzed based on leadership styles, organizational structures, and workplace culture related to psychological safety. The review identified three primary antecedents of psychological safety in high-risk workplaces: (1) inclusive and transformational leadership styles, (2) hierarchical structures and power dynamics, and (3) workplace climate and communication culture. Studies consistently found that inclusive leadership, ethical integrity, and transformational leadership were strongly associated with higher psychological safety levels. While hierarchical structures provided the necessary organization, overly rigid hierarchies often suppressed employee voice and innovation. A workplace culture characterized by trust, transparency, and open communication fosters greater psychological safety and improved teamwork and well-being.

## 1. Introduction

In today’s dynamic and increasingly complex work environments, success is no longer solely dependent on the technical competencies of individuals or the structural efficiency of organizations. Instead, workplace effectiveness is significantly shaped by the quality of interactions among team members, the level of collaboration they engage in, and the overall organizational climate that supports or inhibits open communication [1]. In high-risk occupations such as healthcare and public safety, where rapid decision-making, adaptability, and coordinated teamwork are critical, fostering a psychologically safe work environment is vital to employee well-being, performance, and organizational success [2]. A workplace that cultivates a sense of psychological safety enables employees to fully engage, express their ideas, and contribute to problem-solving without fearing negative repercussions [3].

Psychological safety (PS) in the workplace has gained significant attention in organizational research over the past two decades, particularly concerning its role in enhancing teamwork, reducing workplace stressors, and promoting innovation. Psychological safety was first introduced by Schein and Bennis in 1965 as a means of understanding how organizations could support learning and change by reducing interpersonal fear and risk aversion [2]. Later, Kahn [4] refined the definition by describing psychological safety as a condition in which individuals feel free to express themselves without fearing harm to their self-image, status, or career. Building upon this foundation, Edmondson [5] provided a team-based perspective, defining team psychological safety (TPS) as a shared belief among team members that the team is a safe space for interpersonal risk-taking. This means that within a psychologically safe team, individuals feel comfortable voicing concerns, sharing ideas, admitting mistakes, and seeking feedback without fear of humiliation, punishment, or exclusion.

TPS is particularly critical in team-based work environments, where the success of an organization is increasingly reliant on collective rather than individual efforts. Teams are individuals who work interdependently, dynamically, and adaptively to achieve a shared objective [6]. In high-risk professions, teams ensure safety, efficiency, and performance in environments characterized by uncertainty, complexity, and high-stakes decision-making. However, for teams to function optimally, they must operate in a climate where psychological safety is embedded in the culture, allowing individuals to openly contribute their expertise and insights.

A psychologically safe environment is one where individuals are encouraged and expected to engage in open discussions, challenge existing processes, and voice concerns regarding potential errors or risks. This is particularly essential in high-risk sectors such as healthcare and public safety, where communication failures can lead to dire consequences, including medical mistakes, compromised patient care, safety hazards, and organizational inefficiencies. In contrast, workplaces lacking psychological safety tend to foster silence, disengagement, and a reluctance to report issues, leading to increased errors, decreased morale, and heightened employee burnout.

Creating and sustaining psychological safety is particularly relevant in high-risk workplaces where employees are frequently exposed to stress, unpredictability, and potential harm. In healthcare, law enforcement, firefighting, aviation, emergency response, and military service sectors, employees operate in environments that demand split-second decisions and coordinated teamwork under extreme conditions. In these settings, rigid hierarchies, organizational cultures that discourage speaking up, and punitive approaches to mistakes can significantly impair team performance and employee well-being. Research has shown that fostering TPS in high-risk environments can enhance learning, adaptability, and resilience, ultimately leading to improved outcomes for employees and the individuals they serve [3,5].

In healthcare, patient outcomes depend highly on effective teamwork, communication, and knowledge-sharing among diverse professionals, including doctors, nurses, allied health practitioners, and support staff. Healthcare teams function in high-pressure settings where errors can have severe consequences, making psychological safety a key component of ensuring patient safety and employee well-being. Edmondson [7] identified that healthcare workers in teams with high TPS were likelier to report errors, seek feedback, and share knowledge, leading to improved patient care and reduced medical errors. Additionally, research has demonstrated that hierarchical structures in hospitals often prevent junior staff from voicing concerns, which can result in missed opportunities to avoid adverse events [8]. In contrast, healthcare environments that foster psychological safety encourage staff to communicate openly, challenge problematic decisions, and collaborate more effectively, ultimately improving team cohesion and patient outcomes.

A culture of learning from mistakes rather than punishing them is central to psychological safety in healthcare. Studies have shown that hospitals with high levels of TPS have lower mortality rates, reduced incidence of preventable harm, and higher levels of job satisfaction among healthcare workers [9,10]. Furthermore, initiatives that promote inclusivity and leadership support—such as structured debriefing sessions, patient safety programmes, and leadership training—have been found to increase error reporting rates and improve workplace communication, reinforcing the idea that TPS plays a vital role in shaping healthcare environments.

Public safety professionals, including police officers, firefighters, correctional officers, and paramedics, operate in high-stress environments where they frequently encounter traumatic and life-threatening situations [11]. The high levels of psychological distress associated with these professions make it essential for organizations to foster a work climate that supports team cohesion, trust, and open dialogue. Research has shown that psychological safety in law enforcement is associated with increased willingness to report ethical concerns, improved decision-making under pressure, and enhanced team trust and cooperation [12]. Despite these benefits, public safety organizations’ hierarchical and paramilitary structures often discourage open communication and feedback, as employees may fear retaliation, judgement, or exclusion if they speak out. Studies indicate that fostering TPS in these environments can mitigate these challenges by encouraging employees to share information without fear of repercussions, ultimately improving workplace morale and operational effectiveness [13].

Firefighters and paramedics rely on interdependent teamwork to respond to emergencies quickly and efficiently. In high-risk situations, where individuals must depend on their colleagues for survival and operational success, the ability to speak up, admit uncertainty, and voice concerns can mean the difference between life and death. Training programmes emphasizing inclusive leadership, debriefing sessions, and team-based psychological support have enhanced TPS in emergency response teams, improving coordination, adaptability, and resilience in crises [14].

Psychological safety enables teams to realize their full potential, particularly in high-risk workplace environments where collaboration and trust are paramount. Organizations can create safer, more resilient, and higher-performing teams by fostering a culture prioritizing inclusive leadership, supportive organizational structures, and open communication. This scoping review aims to synthesize the latest research on TPS to provide actionable insights for organizations seeking to enhance psychological safety and improve workplace effectiveness.

Although this review focused primarily on public safety and frontline healthcare sectors, one study conducted in the aviation sector was included due to its significant historical and conceptual relevance to psychological safety research in high-risk industries. The aviation industry has long been recognized as a foundational model for promoting safety culture, communication protocols, and team-based learning—principles that have heavily influenced the development of modern patient safety initiatives in healthcare [3,5]. The Crew Resource Management (CRM) approach pioneered in aviation has been widely adopted across healthcare settings to foster open communication, flatten hierarchical barriers, and encourage speaking-up behaviours critical to patient safety [5]. Because aviation and healthcare share standard features—such as multidisciplinary teamwork, strict protocols, high-consequence environments, and vulnerability to communication failures—the psychological safety mechanisms identified in aviation research are highly transferable and informative for understanding TPS in healthcare and public safety contexts. Therefore, the inclusion of the aviation study was intentional to enrich the review’s conceptual analysis of psychological safety antecedents applicable to other high-risk professional domains.

## 2. Materials and Methods

A preliminary literature search identified a lack of comprehensive data evaluating psychological safety (PS) within the public safety and healthcare sectors, highlighting a significant gap in the literature and motivating the initiation of this scoping review [15]. Researchers frequently employ scoping studies when a research topic remains underdeveloped and its complexity demands a broader exploration [16]. This scoping review will adhere to the framework established by Arksey and O’Malley [17], which includes the following stages: (1) identifying the purpose, (2) identifying relevant studies, (3) selecting studies, (4) charting the data, and (5) collating, summarizing, and disseminating the results.

### 2.1. Study Design

Scoping reviews adopt a broad approach and often incorporate various studies and secondary source information [18]. This scoping review systematically mapped the literature, synthesizing key concepts, theories, and evidence while identifying gaps [17]. The initial step involved conducting a comprehensive search for relevant information to address the primary research question and identifying primary research sources [17,19].

An initial search revealed a notable increase in publications examining psychological safety beginning in 2009, with a significant surge observed after 2015 and a peak of 184 articles published in 2020. To extend the scope of previous reviews on psychological safety, the authors elected to focus on studies published from 2010 onward. This approach facilitates the inclusion of more recent research and enables comparisons between the findings of earlier and current scoping reviews.

### 2.2. Informational Sources

The authors utilized databases to identify high-quality primary resources across public safety, nursing, and medical disciplines. Bibliographic databases provided the beginning search point for the primary and subsequent research questions. The databases searched were both subscription and public databases and included (a) Cumulative Index to Nursing and Allied Health Literature (CINHAL), (b) Medline (Ovid), (c) Educational Resources Information Center (ERIC), (d) Journal Storage (JSTOR), (e) PsycInfo, (f) Business Source Ultimate, (g) Sociological Abstract. The authors generated alerts for each database and monitored them weekly for newly published studies until April 2025. Databases were accessed through the Thompson Rivers University Discovery search system. The authors manually searched for additional articles using the Data Mining method and included those that met the inclusion criteria outlined in this scoping review [20].

### 2.3. Search Strategy

References cited in the first primary resource search were used to expand the search further [19]. A search using keywords and descriptors was carried out, and a thorough review of the available literature was conducted. A research librarian was utilized to aid in the identification of relevant search terms. The research tool SPIDER (Sample, Phenomenon of Interest, Design, Evaluation, and Research type) is utilized as the framework for the strategy employed in the search (see Table 1). The SPIDER tool defines this scoping review and its employed search strategy.

### 2.4. Selection of Studies

The search strategy focused on databases containing peer-reviewed systematic or empirical articles as a standard. The screening process involved reviewing titles and abstracts, with duplicate articles excluded. Articles that satisfied the inclusion criteria underwent full-text review for comprehensive analysis. The article selection process and inclusion and exclusion parameters are detailed in a Preferred Reporting Items for Systematic Reviews and Meta-Analyses (PRISMA) extensions for Scoping Reviews flow diagram (Figure 1). Additionally, the authors employed the ancestry approach to identify and reference relevant sources from the bibliographies of included articles.

The initial search yielded 1119 articles, with 303 remaining after removing duplicates. To enhance consistency among reviewers, authors independently reviewed the titles and abstracts of 132 articles, discussed their findings, and refined the screening and data extraction criteria. The authors discussed conflicts and refined the inclusion criteria before reviewing the remaining 171 titles and abstracts [17]. Subsequently, the authors conducted a hand search of references from the included articles to identify additional studies meeting the inclusion criteria. A third reviewer (RS or GA) resolved any conflicts during the article selection process. Following a comprehensive review, 16 articles were ultimately included in this study.

### 2.5. Inclusion and Exclusion Criteria

To be included in this review, articles had to meet the following inclusion criteria: (1) published between January 2010 and January 2025; (2) peer-reviewed and reporting primary empirical studies (excluding books, conference papers, commentaries, theoretical papers, theses, systematic reviews, and essays); (3) examined antecedents to team psychological safety (TPS); (4) measured TPS using a validated questionnaire; (5) written in English; (6) focused on a population of adult workers; and (7) conducted in high-risk workplace environments (see Table 2).

Given the contextual specificity of TPS, the review excluded studies conducted in collectivist countries. Collectivism emphasizes interdependence among individuals within a collective, contrasting with individualism, where decisions and actions prioritize personal autonomy and individual benefit. In collectivist cultures, individuals are more likely to prioritize team goals and act for the collective good, reducing the variability in TPS based on individual behaviours. Therefore, this review focused on countries with a predominantly individualistic culture. Included studies were conducted in Australia, Austria, Belgium, Bulgaria, Canada, Croatia, Cyprus, Czech Republic, Denmark, Estonia, Finland, France, Germany, Greece, Hungary, Iceland, Ireland, Italy, Latvia, Liechtenstein, Lithuania, Luxembourg, Malta, Netherlands, New Zealand, Norway, Poland, Portugal, Romania, Slovakia, Slovenia, Spain, Sweden, Switzerland, the United Kingdom (England, Northern Ireland, Scotland, Wales), and the United States.

Research investigating the implementation of peer support initiatives and crisis-oriented psychological interventions aimed at mitigating the sequelae associated with exposure to potentially psychologically traumatic events (PPTE) among adults (aged 18 and older) employed as public safety personnel (PSP) and frontline healthcare professionals (FHP) were eligible for inclusion. Eligible PSP roles included border services officers, correctional officers, communications personnel (e.g., dispatch operators and 911 operators), firefighters, paramedics, and police officers. FHP roles encompassed nurses, professionals working in emergency departments, trauma centres, surgical teams, social workers, and counsellors [22].

Given the contextual specificity of team psychological safety (TPS), this review excluded studies from collectivist countries. Collectivist cultures emphasize interdependence and group harmony, often discouraging individuals from speaking up or challenging authority [20,23]. In contrast, individualistic societies prioritize autonomy and personal voice, making them more conceptually aligned with the constructs underpinning TPS. Research demonstrates that cultural values, such as individualism–collectivism and power distance, significantly influence behaviours like speaking up and participatory decision-making—core aspects of TPS [20]. However, excluding collectivist contexts does not suggest culture overrides the effects of leadership or organizational interventions. Evidence shows that, while approaches may require cultural adaptation, interventions can enhance psychological safety across diverse settings [24]. Future research should explore culturally responsive strategies to promote TPS globally.

## 3. Results

This scoping review systematically explored research on antecedents of team psychological safety (TPS) in high-risk workplace environments, focusing on healthcare and public safety sectors. Through a rigorous selection process, 16 studies were identified, analyzed, and synthesized to assess the key determinants of TPS. The findings from this scoping review highlight several key factors influencing TPS in high-risk occupations. Leadership styles emphasizing inclusivity, ethical integrity, and transformational behaviours were frequently associated with TPS, suggesting that organizations should prioritize leadership training and development programmes that reinforce these qualities. The research also indicated that hierarchical structures, while necessary for maintaining order, require balance to ensure they do not suppress employee voice. Lastly, cultivating a caring workplace climate and fostering open communication is essential for enhancing psychological safety, ultimately leading to improved teamwork, employee well-being, and organizational effectiveness.

### 3.1. Study Selection Summary

Sixteen articles met all predefined selection criteria. The studies span six countries, with North America and Europe contributing most of the research. Healthcare-related studies accounted for a significant portion of the included literature, followed by public safety occupations, such as law enforcement and aviation. The PRISMA extension for scoping reviews framework guided the selection process, ensuring a structured and transparent approach.

### 3.2. Data Charting

The authors employed predetermined data charting forms based on Arksey and O’Malley’s [17] framework to capture relevant information from the literature systematically. To facilitate data extraction, the authors developed a comprehensive table capturing essential aspects of the evidence, including the author, year, location, study site, purpose, participants, study design, statistical methods, TPS as an outcome or moderator, measurement of TPS, reliability of the TPS measure, and study results. A summary of the findings is presented in Table 3.

### 3.3. Characteristics of Included Studies

The reviewed studies encompassed a diverse array of methodological designs, including cross-sectional surveys (n = 11), longitudinal studies (n = 3), and field experiments (n = 2). Qualitative, quantitative, and mixed-methods approaches were employed, with survey-based assessments being the most common methodology. Most studies relied on Edmondson’s [5] psychological safety scale, with adaptations in various workplace settings. The studies included high-risk workplaces, hospitals, law enforcement agencies, and aviation teams. The populations studied included healthcare professionals (nurses, physicians, and allied health workers), law enforcement personnel, and airline crew members. Sample sizes varied significantly, with the most minor study including 34 participants and the most extensive study involving 1490 individuals. The geographic distribution of studies revealed a research concentration in Western countries, suggesting potential regional biases in understanding TPS antecedents.

While this scoping review identified inclusive leadership, hierarchical structure, and workplace climate as key antecedents to psychological safety (PS) in high-risk occupations, a significant limitation is the lack of consistent quantification of these relationships. Most studies used cross-sectional designs and qualitative analyses, limiting causal inference and effect size estimation. Emerging empirical work has begun addressing this gap: inclusive leadership shows moderate to strong associations with PS [13,25], and transformational leadership demonstrates a robust impact [26]. Leadership integrity also positively correlates with PS outcomes [10]. The role of hierarchy appears bidirectional—procedural fairness enhances PS [11], while rigid hierarchies under high strain diminish it [25]. A caring climate further promotes PS [27], with communication quality mediating these effects [28]. However, variability across contexts, samples, and measures limits generalizability. Future longitudinal and experimental studies are needed to clarify causal pathways and strengthen intervention development.

### 3.4. Thematic Analysis and Key Findings

A narrative analysis and synthesis of the data following the approach outlined by Popay et al. [24] was used to identify and summarize the current literature on antecedents to TPS in high-risk workplace environments. Several key factors contribute to fostering TPS in high-risk work environments, including (1) Leadership Style; leadership plays a crucial role in shaping psychological safety. Studies have shown that inclusive, transformational, and ethical leadership styles are associated with higher TPS levels [23,25]. Leaders who encourage employee input, value diverse perspectives, and provide emotional support create environments where individuals feel safe to engage in open discussions. (2) Hierarchical Structures and Organizational Culture: while hierarchy is necessary for maintaining order and accountability in complex organizations, excessive power differentials can suppress psychological safety. Workplaces that balance structure with inclusivity—promoting fairness, procedural justice, and transparent decision-making—are more likely to foster a culture of psychological safety [12]. (3) Workplace Climate and Communication: organizations that emphasize open dialogue, trust, and a learning culture rather than punishment tend to exhibit higher levels of psychological safety. Effective feedback mechanisms, debriefing sessions, and conflict resolution strategies contribute to fostering TPS [7].

### 3.5. Leadership as a Determinant of TPS

Leadership emerged as the most frequently examined antecedent, with all 16 studies highlighting its role in fostering TPS. Leaders who exhibited inclusive, transformational, ethical, and change-oriented leadership behaviours consistently contributed to higher psychological safety among team members.

#### 3.5.1. Inclusive Leadership and TPS

Inclusive leadership, defined by leader accessibility, open communication, and encouragement of diverse perspectives, was found to be a significant predictor of TPS in multiple studies [13,29,30,31,32]. For instance, Diabes et al. [31] reported that leader inclusiveness significantly predicted TPS (β = 0.32; *p* < 0.05) among healthcare teams, suggesting that when leaders actively seek input and acknowledge team contributions, employees feel safer engaging in workplace discussions. Similarly, Hassan and Jiang [13] found that inclusive leadership positively correlated with workgroup psychological safety (β = 0.62, *p* < 0.01) within law enforcement teams, highlighting the importance of leadership in shaping perceptions of psychological safety in hierarchical organizations. Leaders who actively sought employee input encouraged participation and demonstrated an openness to diverse perspectives significantly enhanced TPS [13,30,31]. Inclusive leadership was associated with increased employee engagement, improved team performance, and a greater willingness to report concerns.

#### 3.5.2. Interprofessional Teams and TPS

Interprofessional teams, particularly in high-risk sectors such as healthcare and aviation, exhibited distinct leadership dynamics influencing TPS. In healthcare, leadership was often shared between physicians and nurse managers, each fulfilling unique roles. Physician leadership style correlated more strongly with TPS (β = 0.38, *p* < 0.10), potentially due to their authoritative status and perceived expertise [33]. Nurse managers contributed significantly to interdependence and team cohesion by focusing on patient-centred care. In aviation, interprofessional teams operate within a culture that emphasizes open communication and reduces authoritarian leadership styles. A study of 1490 airline crew members found that leader inclusiveness significantly predicted TPS (β = 0.57, *p* < 0.001 for flight attendants; β = 0.59, *p* < 0.001 for first officers) [30]. However, while TPS mediated speaking-up behaviours, it did not mediate communication between teams, indicating potential barriers in inter-team interactions.

#### 3.5.3. Leadership Integrity and TPS

Leadership integrity also played a crucial role in fostering TPS. Studies on leader behavioural integrity demonstrated that when leaders’ actions aligned with their stated values, TPS increased, improving safety compliance and error reporting [9,10]. For example, Leroy et al. [10] found a positive association between leader behavioural integrity and TPS (β = 0.34, *p* = 0.01), emphasizing that consistency between leadership rhetoric and practice enhances employees’ willingness to voice concerns. Leaders who inspired their teams through vision, motivation, and empowerment contributed to higher levels of TPS. Raes et al. [34] found a significant positive correlation between transformational leadership and TPS (β = 0.70, *p* < 0.001).

#### 3.5.4. Types of Leadership

Transformational leadership, which emphasizes motivation, empowerment, and vision, was similarly associated with TPS. Raes et al. [34] found that transformational leadership significantly predicted TPS (β = 0.70, *p* < 0.001), with TPS mediating the relationship between leadership and team learning behaviours. Moreover, studies comparing leadership styles suggested that transformational leadership was more effective in fostering TPS than laissez-faire leadership (β = 0.39, *p* < 0.05) [34]. Studies highlighted those leaders who demonstrated ethical integrity and aligned their actions with organizational values fostered a culture of trust and psychological safety [8,10]. A positive relationship between ethical leadership and TPS was observed.

### 3.6. Organizational Hierarchy and Structure

The impact of hierarchy and structure on TPS varied across studies, with some findings suggesting that structured environments could facilitate TPS under certain conditions. In high risk or emergency situations a strong hierarchy is important and effective when quick decisions need to me made and time is not available for robust discussions and sharing of opinions [12]. Studies in law enforcement revealed that perceptions of distributive justice and procedural fairness significantly predicted TPS (β = 0.09, *p* < 0.05 and β = 0.43, *p* < 0.001, respectively) [12]. Officers who perceived just treatment were likelier to engage in open discussions and knowledge sharing. These findings suggest that hierarchical organizations can support TPS when decision-making processes are perceived as just and transparent.

Conversely, overly rigid structures were found to inhibit open communication and suppress psychological safety. In healthcare, excessive hierarchical structures created barriers to open communication. Diabes et al. [31] observed that job strain negatively correlated with TPS (β = −0.07; 95% CI, −0.13 to −0.02), highlighting that during periods of excessive workload pressures strict hierarchical protocols could discourage employees from speaking up. Similarly, aviation research indicated that team status differentials negatively influenced voice behaviours unless leaders actively fostered an inclusive and psychologically safe culture [30]. In aviation, team status differences inhibited voice behaviours unless mitigated by an inclusive leadership approach [30]. Teams with high-status differentials demonstrated lower TPS levels than those with more egalitarian structures.

### 3.7. Workplace Climate and Communication Culture

The role of workplace climate and organizational culture was another prominent theme in the reviewed literature. Organizational climate played a pivotal role in shaping TPS. Studies found trust, transparency, and open communication strongly linked to increased TPS.

Studies consistently found that a caring climate—prioritizing interpersonal respect, support, and open communication—was associated with higher TPS levels [9,26,27]. Kruzich et al. [26] demonstrated that a caring work climate significantly predicted TPS (β = 0.53, *p* < 0.001) and was negatively associated with emotional exhaustion (β = −0.42, *p* < 0.001). A caring workplace environment was associated with lower emotional exhaustion and higher psychological safety [26,27]. Healthcare professionals working in a supportive climate exhibited greater resilience and teamwork effectiveness. These findings suggest that organizations prioritizing employee well-being and supportive leadership can enhance psychological safety, improve job satisfaction, and reduce burnout.

Communication emerged as a central mechanism through which TPS was established and maintained. Studies highlighted that transparent and open communication channels facilitated trust, knowledge sharing, and team cohesion [9,28]. Effective communication mechanisms, including structured debriefings and routine feedback sessions, fostered TPS [9,26]. In contrast, environments characterized by defensive or punitive communication patterns demonstrated lower levels of TPS, with employees exhibiting reluctance to voice concerns due to fear of negative repercussions [25,27].

**Table 3 ijerph-22-00820-t003:** Summary of findings.

Author, Year	Study Location	Study Design and Level	PS Safety Measure	Alpha	PS Antecedent	Results
**Alingh et al. 2018 [9]**	Netherlands Hospital	Cross-Sectional Team Level	Edmondson 1999-7pt [5]	0.77	Leadership	Commitment-based safety management, compared to control-based approaches, positively influences nurses’ perceptions of safety and psychological safety, fostering healthier attitudes toward speaking up about patient concerns. A significant association was found between commitment-based safety management and team psychological safety (β = 0.36; *p* < 0.01), suggesting that higher perceived commitment enhances nurses’ willingness to take interpersonal risks. Control-based safety management was also positively associated with the overall safety climate.
**Appelbaum et al. 2016 [25]**	USA Hospital	Cross-Sectional Individual Level	Edmondson 1999-5pt	N/A	Leadership	Psychological safety was positively associated with leadership inclusiveness and negatively associated with power distance, highlighting the value of inclusive leadership in promoting event reporting and fostering a safer, more transparent workplace. This study found that perceived power distance (β = 0.26; SE = 0.06; 95% CI [0.37, 0.15]; *p* < 0.001) and leader inclusiveness (β = 0.51; SE = 0.07; 95% CI [0.38, 0.65]; *p* < 0.001) significantly predict psychological safety.
**Bienefeild & Grote 2014 [29]**	Switzerland Airlines	Cross-Sectional Individual Level	Baer and Frese 2003-7pt (organization)	Sample 1 = 0.78; Sample 2 = 0.77; Sample 3 = 0.72	Leadership	This study examined whether mechanisms promoting speaking up within a single team extend across multiple teams. Findings showed that subjective status (Sample 1: β = 0.10, *p* < 0.001; Sample 2: β = 0.12, *p* < 0.001) and leader inclusiveness (Sample 1: β = 0.57, *p* < 0.001; Sample 2: β = 0.59, *p* < 0.001) significantly predict psychological safety within teams. Across teams, within-team psychological safety significantly promotes speaking up (β = 3.50, *p* = 0.01) and reduces the influence of subjective status (β = 0.98, *p* = 0.02). These results underscore the role of psychological safety in encouraging voice both within and across teams.
**Diabes et al. 2021 [30]**	USA Hospital	Longitudinal Individual Level	Edmondson 1999-5pt	0.69	Leadership	An observational study across six hospitals and twelve ICUs found that leader inclusiveness positively predicts psychological safety (adjusted β = 0.32; 95% CI, 0.24–0.41), while perceived job strain negatively predicts it (adjusted β = −0.07; 95% CI, −0.13 to −0.02). Role clarity initially correlated with psychological safety and teamwork but lost significance after adjustment, and leader familiarity showed no significant association. These findings highlight modifiable factors influencing psychological safety in ICUs, though implementation does not always align with evidence-based practises.
**Halbesleben et al. 2013 [8]**	USA Hospital	Retrospective Individual Level	Edmondson 1999-5pt	0.9	Leadership	Expanding research on behavioural integrity and occupational safety in healthcare, this study found that safety-related behavioural integrity is positively associated with psychological safety toward one’s supervisor (β = 0.32, *p* ≤ 0.05). Higher levels of behavioural integrity were also linked to reduced frequency and severity of occupational injuries, along with greater psychological safety and safety compliance. The findings suggest that psychological safety improves when leaders consistently model safety values, and that nurses in psychologically safe environments are more likely to report safety concerns.
**Hassan & Jiang 2021 [13]**	USA Law Enforcement	Cross-Sectional Team Level	Edmondson 1999-5pt	0.97	Leadership	Researchers conducted a study following reports of police brutality online to examine how law enforcement managers’ participation in workshops designed to cultivate learning could enhance performance. The findings indicate that one standard deviation (SD) increase in inclusive leadership behaviour corresponds to a 0.62 SD increase in workgroup psychological safety. Inclusive leadership behaviour demonstrates a significant positive relationship with workgroup psychological safety (β = 0.62, *p* < 0.01). The study concludes that the demonstration of inclusive leadership is positively associated with fostering psychological safety within workgroups.
**Kolbe et al. 2013 [31]**	Switzerland Hospital	Longitudinal pre-post interventionTeam Level	Edmondson 1999-5pt	0.69	Leadership	To enhance patient safety and develop a more effective simulation evaluation tool, researchers designed TeamGAINS to improve debriefing approaches. This study evaluated the effectiveness of three different TeamGAINS debriefing techniques in simulation scenarios. The findings indicate that mean scores of psychological safety significantly increased from *M* = 3.36, *SD* = 0.63 to *M* = 3.48, *SD* = 0.54 (*p* = 0.028). At the same time, leader inclusiveness significantly increased from *M* = 3.21, *SD* = 0.68 to *M* = 3.33, *SD* = 0.56 (*p* = 0.048) after debriefing sessions. These results suggest that implementing the TeamGAINS debriefing tool enhances psychological safety and leader inclusiveness in subsequent simulation scenarios.
**Kruzich et al., 2014 [34]**	USA Hospital	Cross-Sectional Individual Level	Edmondson 1996-4pt	0.67	Climate	This study examined how caring climate, psychological safety, and empowerment influence staff retention in child welfare organizations. A caring climate positively predicted psychological safety (b = 0.53, t = 13.73, *p* < 0.001), which in turn negatively predicted emotional exhaustion (b = −0.42, t = −7.27, *p* < 0.001). Empowerment moderated this relationship, with lower empowerment strengthening the link between caring climate and psychological safety. Moderated mediation analysis confirmed that the indirect effect of caring climate on emotional exhaustion through psychological safety varied by empowerment level. These findings underscore the critical role of fostering a caring climate and empowerment to reduce emotional exhaustion and enhance workforce stability in high-stress environments.
**Leroy et al., 2012 [10]**	Belgium Hospital	Cross-Sectional Team Level	Edmondson 1999-5pt	0.8	Leadership	Examining the relationship between leadership behaviour and workplace safety, researchers found that leader behavioural integrity for safety is positively associated with team psychological safety (β = 0.34, *p* = 0.01), suggesting that leaders who consistently uphold safety values create environments where employees feel safe to report mistakes. The findings also show that prioritizing safety and fostering psychological safety significantly predicts the frequency of reported treatment errors in hospital settings, highlighting the critical role of leadership integrity in improving safety compliance and error reporting.
**Liu et al., 2018 [28]**	USA Health Organization	Cross-Sectional Organization Level	Edmondson 1999-7pt	0.78	Leadership	Investigating the impact of authentic leadership on workplace dynamics, researchers found a significant positive relationship between authentic leadership and psychological safety (r = 0.46, *p* < 0.01), indicating that authentic leaders foster environments where employees feel safe to express ideas and take risks. Supervisor identification mediated this relationship, with stronger identification enhancing psychological safety and, in turn, promoting greater job engagement.
**Ortega et al., 2014 [35]**	Spain Hospital	Cross-Sectional Team Level	Edmondson 1999-5pt	0.67	Leadership	This study examined how psychological safety and change-oriented leadership influence team performance and learning in healthcare settings. Change-oriented leadership was significantly associated with psychological safety (β = 0.52, *p* < 0.001), directly enhancing team learning and indirectly fostering it through increased psychological safety. These findings highlight the importance of developing change-oriented leadership to strengthen team effectiveness and improve patient care quality.
**Raes et al., 2013 [33]**	Netherlands Hospital	Cross-Sectional Team Level	Edmondson 1999-7pt	0.71	Leadership	Leadership style was examined to compare the effects of transformational and laissez-faire approaches on team psychological safety and learning behaviours. Transformational leadership significantly predicted psychological safety (β = 0.70, t = 4.42, *p* < 0.001), while laissez-faire leadership demonstrated a weaker positive relationship (β = 0.39, t = 2.34, *p* < 0.05). Findings suggest that transformational leadership fosters psychological safety prior to social cohesion, whereas laissez-faire leadership prioritizes cohesion first. These results highlight the stronger role of transformational leadership in promoting team learning through psychological safety.
**Ridley et al., 2021 [14]**	USA Medical Center	Longitudinal Team Level	Edmondson 1999-5pt		Climate/Culture	Implementation of a structured teamwork training programme in a cardiothoracic operating room unit resulted in an increase in positive psychological safety from 78.1% to 88.2% over 12 months, although this change was not statistically significant (difference = 10.1%; 95% CI, −2.4% to 23.4%; *p* = 0.122). Positive psychological safety was reported by 57 of 73 staff at baseline and 60 of 68 after training. However, significant improvements were observed in components of psychological safety, with positive perceptions of teamwork increasing from 82.2% to 90.9% (*p* < 0.001) and speaking up rising from 87.4% to 93.9% (*p* < 0.001). These findings suggest that structured training can meaningfully enhance teamwork, psychological safety, communication, and patient outcomes.
**Sumanth et al., 2024 [36]**	USA Fire Organization	Field studies Individual Level	Harmans single factor test		Leadership	Exploring factors that influence employees’ willingness to report counterproductive work behaviours (CWBs), this research tested a model across three field studies in standard and extreme work environments. Findings show that ethical leadership enhances moral potency, increasing peer reporting, while the effect of psychological safety varies by context. In standard environments, psychological safety strengthens the link between moral potency and reporting intentions, whereas in high-risk settings, it weakens it. Specifically, higher psychological safety amplified reporting among firefighters with lower extreme contextual exposure (t = 1.89, *p* = 0.063) but had little impact under higher exposure (t = −0.82, *p* = 0.417). These results illustrate how self-regulatory capacity and contextual factors shape ethical action in organizations.
**Wholey et al., 2014 [32]**	USA Medical Center	Cross-Sectional Team Level	Author Designed Survey	PSYC Safety: (alpha = 0.85; ICC = 0.10);	Leadership	Leadership’s role in shaping psychological safety, interdependence, and team coordination in healthcare was explored. Findings show that physician leadership significantly enhances psychological safety (β = 0.38, t = 4.36), respect (β = 0.39, *p* < 0.1), and shared goals (β = 0.40, *p* < 0.1), fostering a climate where team members feel safe to speak up. Nurse leadership strengthens interdependence by structuring workflows and managing patient interactions, promoting seamless coordination. Additionally, group support reinforces psychological safety and team climate. Together, physician and nurse leadership foster collaboration, learning, and team preparedness, emphasizing the critical balance needed to optimize healthcare teamwork.
**Workman-Stark, 2020 [12]**	Canada Police Organization	Cross-Sectional Team Level	Edmondson 1999-7pt	0.74	Hierarchy/Structure	Identifying the role of perceived fairness in decision-making and outcomes, this study found that fair treatment significantly enhances psychological safety among police personnel, boosting organizational identification and job engagement. Distributive justice (α = 0.65, *p* < 0.05) and procedural justice (α = 0.52, *p* < 0.05) were both positively associated with psychological safety. Psychological safety also mediated the relationship between justice perceptions and engagement, with significant indirect effects for distributive justice (β = 0.09, *p* < 0.05, SE = 0.04) and procedural justice (β = 0.43, *p* < 0.001, SE = 0.06). Notably, distributive justice played a particularly strong role in fostering organizational commitment. These findings emphasize the importance of cultivating fairness and psychological safety to strengthen workforce engagement in law enforcement.

## 4. Discussion

This scoping review identified several factors that positively contribute to team psychological safety (TPS) in high-risk occupations, including inclusive leadership styles, hierarchical structures, and a workplace culture characterized by care and support. While these antecedents represent broad conceptual domains, several specific themes emerged across TPS-related factors. These antecedents enhance TPS by fostering employees’ sense of value, belonging, and trust through effective communication, inclusivity, consistent messaging, and ethical conduct within the organization or team. Furthermore, cultivating a culture of trust—both in leadership and among coworkers—may be a critical determinant in establishing a psychologically safe work environment. The following sections provide a detailed discussion of the identified antecedents: leadership, hierarchy and structure, and organizational climate and culture.

### 4.1. Leadership

Consistent with previous reviews, researchers have identified certain leadership styles and behaviours can positively influences team psychological safety (TPS) in high-risk workplaces [31,37]. Edmondson [7] observed that leaders who adopt inclusive leadership styles actively listen to employees, consider their perspectives and ideas and express appreciation for their contributions, fostering a sense of value among employees. Recognizing employees as valuable assets and resources—and effectively communicating this recognition—is critical in cultivating a psychologically safe work environment.

Leadership and management, though closely related, serve distinct roles in an organization. Management focuses on maintaining stability and efficiency through structured planning, goal setting, and resource allocation, ensuring that operational processes run smoothly [13,36]. Leadership, in contrast, is vision-driven, emphasizing innovation, inclusivity, and cultural transformation [38]. While both are essential, an organization that leans too heavily on structure risks employee disengagement, diminished creativity, and higher turnover [13]. Conversely, a workplace prioritizing vision and inclusivity without effective management can struggle with inefficiency, inconsistent outcomes, and organizational instability [36].

Leadership and management must work together to create a workplace where psychological safety and operational success coexist. Managers who balance structure with flexibility, encourage open dialogue, and foster a culture of trust can reinforce psychological safety while maintaining efficiency [23]. Effective leaders inspire teams, empower employees, and challenge existing norms, creating an environment where individuals feel safe to voice concerns, admit mistakes, and contribute ideas without fear of retribution [5,37]. This balance is particularly critical in high-risk industries, where the consequences of silence, disengagement, or poor decision-making can be severe [25,34].

The most effective organizations recognize that leadership is not limited to executive roles—managers at all levels can adopt leadership-oriented behaviours to enhance psychological safety [39]. Training programmes incorporating active listening, emotional intelligence, and participatory decision-making equip managers with the tools to create a culture of psychological safety while ensuring operational stability [23]. Research consistently shows that organizations that invest in leadership development see improvements in employee engagement, collaboration, and overall performance [10]. By integrating leadership principles into management practises, organizations can foster an environment where employees feel secure and motivated, ultimately enhancing innovation, teamwork, and long-term success [7,29].

Leadership, likely due to its human-centred approach, emerged as the most frequently examined antecedent to TPS in the reviewed literature, with 16 studies analyzing its impact. Workplace leadership influences employees at multiple levels, from organizational leadership to team supervisors and individual team members [39]. Leaders at any level shape employee TPS by modelling behaviours, fostering organizational culture, developing policies and practises, and maintaining transparent and effective communication [33]. Leadership styles vary based on the distribution of decision-making authority and communication strategies, leading to diverse effects on organizational employees [29,33]. Various leadership styles are discussed in the literature concerning TPS, such as authentic, inclusive, and transformational, and each contributes to continuing assorted outcomes and consequences on employees.

### 4.2. Inclusive Leadership

Inclusive leaders actively involve others in discussions and decision-making processes from which they might otherwise be excluded, demonstrating openness, accessibility, and availability [1]. Inclusive leadership also involves assembling diverse teams, valuing and leveraging diversity, maintaining an open perspective, building meaningful relationships, and fostering a collaborative work environment. Additionally, inclusive leaders facilitate teamwork, listen attentively, learn from others, treat individuals equitably, cultivate a welcoming and constructive atmosphere, encourage employees to contribute ideas, instil a sense of empowerment, and establish trust within the organization [40]. Five studies in this review examined the relationship between leader inclusiveness and TPS in high-risk occupations [13,25,29,30,31]. Inclusive leaders should also recognize and effectively navigate issues of power, privilege, and marginalization to ensure that all team members feel genuinely valued and heard; however, these themes did not prominently emerge in the reviewed literature, highlighting a gap in research on how inclusive leadership can address systemic inequities and foster psychological safety for underrepresented groups.

Hospital-based medical teams are inherently diverse and interprofessional, comprising individuals with varying skill sets, scopes of practice, and levels of authority [33]. The effectiveness of these assorted teams is critical, as patient well-being depends on their performance. Cross-sectional studies have examined the relationship between inclusive leadership and TPS within high-stress occupations [12,29,31].

In the first study, intensive care unit teams completed surveys to assess the relationship between inclusive leadership and TPS from an individual perspective. Findings indicated that higher levels of leader inclusiveness corresponded with increased TPS among team members (β = 0.32; *p* < 0.05). Additionally, TPS was positively associated with an enhanced perception of teamwork, suggesting that when leaders adopt inclusive behaviours—such as pausing discussions to allow input, actively seeking team members’ contributions, explicitly explaining key decisions, and acknowledging uncertainty to encourage open dialogue—they create a safer environment for staff to voice concerns [31]. Correspondingly, a study on resident physicians found that leader inclusiveness was significantly associated with higher TPS (β = 0.51; 95% CI 0.38 to 0.65; *p* < 0.001) and increased resident intentions to report adverse events [29]. The authors propose that because organizational culture shapes the willingness to report adverse events, individuals and institutions share responsibility for reporting adverse events [29]. Moreover, post-training debriefings may serve as a mechanism for fostering inclusivity and enhancing TPS within interprofessional teams. Hospital staff who participated in such debriefings reported significant increases in perceptions of leader inclusiveness (*p* = 0.03) and TPS (*p* = 0.05) [29].

The third study highlights the influence of workgroup learning in high-stress occupations by enhancing perceptions of teamwork and strengthening TPS. The study examining law enforcement managers and subordinates explored the relationship between inclusive leadership, workgroup learning, and workgroup performance. Law enforcement officers operate with substantial autonomy and discretion, with their behaviours strongly shaped by training and the distinct cultural norms of policing [12]. Due to these factors, officers often resist organizational change [12,13]. Linear regression analysis of survey data revealed that inclusive leadership behaviours were positively and significantly associated with workgroup TPS (β = 0.62, *p* < 0.01), which in turn contributed to significant improvements in workgroup learning and performance [13]. Implementing a more inclusive leadership approach may mitigate this resistance by fostering open communication within the organization and counterbalancing the hierarchical occupational culture, which traditionally prioritizes collective decision-making over individual perspectives [13].

Participative Leadership. Participative leadership behaviour, a subset of inclusive leadership, involves soliciting employee input and consultation in decision-making processes [41]. Similarly, perceived supervisor support reflects the extent to which supervisors recognize and value employees’ contributions while demonstrating concern for their well-being [42]. A study investigating the impact of supportive and empowering leadership on employee perceptions of supervisor support and its relationship with TPS among child welfare workers found a significant positive association. The results indicated that supportive and empowering leadership significantly predicted TPS (β = 0.52, *p* < 0.001) [26]. A participative leadership framework can substantially reduce employee turnover rates and enhance morale and workplace conditions [41]. This improvement occurs as employees perceive their contributions as valuable and instrumental in driving organizational change [26].

### 4.3. Interprofessional Teams

Interprofessional teams are inherently complex, often comprising multiple leaders with distinct yet complementary roles and functions. For instance, leadership is typically shared between a physician and a nurse manager within medical teams, each contributing uniquely to team dynamics [33,37]. The physician embodies status leadership, leveraging their advanced medical expertise to focus on patient diagnosis and treatment [37]. In contrast, the nurse manager assumes a process-oriented leadership role, prioritizing patient-centred care and monitoring patient status [33]. Through varied mechanisms, these differentiated leadership roles influence healthcare team members’ transactive processes. Empirical findings indicate that both nurse (r = 0.54) and physician (r = 0.39) leadership exhibit positive correlations with interdependence and key team climate factors, including TPS (r = 0.32 and 0.43, respectively), respect (r = 0.47 and 0.54, respectively), and shared goals (r = 0.46 and 0.58, respectively). Notably, only physician leadership demonstrated a statistically significant positive relationship with TPS (β = 0.38, *p* < 0.10) [33]. The underlying reasons for physician leadership’s comparatively more significant effect on TPS remain unclear. One potential explanation is that the physician’s authoritative status as a medical expert may enhance the trust and confidence of team members, fostering an environment in which they feel more assured of achieving positive patient outcomes [8,33].

Like healthcare professionals and law enforcement personnel, airline employees frequently encounter high-risk situations in which teamwork and effective communication are critical. Utilizing an interprofessional leadership aviation team, each member contributes specialized knowledge, experience, and expertise to ensure optimal aviation safety within a culture that prioritizes open, trustworthy communication while minimizing authoritarian practises [43]. This shift is associated with reduced errors, improved task delegation, increased employee morale, and decreased turnover rates [30,43]. Benefield and Grote [30] conducted a study involving 1490 aircrew members from a European airline to investigate the mechanisms underlying speaking up within and across teams. Their findings revealed that leader inclusiveness was a significant predictor of transactive multiteam systems within both a cohort of flight attendants (β = 0.57, *p* < 0.001) and a cohort of first officers (β = 0.59, *p* < 0.001) at comparable levels. Furthermore, transactive multiteam systems functioned as a mediator between inclusive leadership and speaking up within teams. However, when examining inter-team communication, transactive multiteam systems did not mediate the relationship between inclusive leadership and speaking up [30].

### 4.4. Leadership Integrity

Safety initiatives are of critical importance in healthcare. Establishing a psychologically safe environment where healthcare professionals feel comfortable acknowledging errors has enhanced error reporting and reduced medical mistakes [9,10]. To cultivate such an environment, leadership qualities such as role modelling and trust-building may be essential [8,9,10].

Leader behavioural integrity, first introduced by Simons [44], refers to employees’ perception of how much a leader’s actions align with their stated values and is directly associated with employee trust. Empirical evidence suggests that when a head nurse demonstrates behavioural integrity in prioritizing safety, team perceptions of safety importance (β = 0.37, *p* < 0.05), and transactive memory systems (β = 0.34, *p* < 0.05) are positively influenced [10]. Furthermore, this alignment is associated with increased error reporting and reduced error occurrences. These findings suggest that when head nurses internalize and consistently uphold safety values, they send unambiguous signals to their teams, reinforcing the prioritization of safety over competing demands.

Leader behavioural integrity aligns conceptually with control-based and commitment-based safety management approaches, as leaders emphasize the importance of safety, enforce adherence to safety protocols, and model behaviours that demonstrate their commitment to safety practises [9]. In a cohort of nurses, control-based safety management demonstrated a positive relationship with the climate for patient safety but was negatively associated with psychological safety (PS). Conversely, nurse managers were committed to safety management through a framework of integrity leadership, positively correlated with PS (β = 0.36, *p* < 0.01). Notably, both leadership styles independently exhibited a positive association with speaking up about safety concerns [9].

A similar pattern emerges in the relationship between leader behavioural integrity and occupational injuries, wherein subordinate PS toward nurse supervisors mediates (β = 0.32, *p* ≤ 0.05) the effect of leader behaviour on occupational safety outcomes. This mediation reduces the frequency and severity of workplace injuries and increases injury reporting [8]. These findings suggest that leadership approaches emphasizing integrity and commitment to safety can foster a culture of openness, enhance adherence to safety practises, and ultimately improve patient and occupational safety outcomes.

#### 4.4.1. Authentic Leadership

An authentic leader fosters trust by demonstrating self-awareness, understanding their internal thoughts, beliefs, and emotions, and adhering to an internalized moral compass [45]. Additionally, authentic leaders consistently exhibit supportive and transparent behaviours in their interactions with employees [35,45]. Researchers believe these authentic leadership qualities strengthen trust within professional relationships and reduce workplace uncertainty. In the context of transactive memory systems, studies have shown that authentic leadership positively correlates with TPS and subordinates’ identification with their supervisor (r = 0.46, *p* < 0.001). This finding suggests that employees’ attachment to their supervisors is associated with authentic leadership and psychological safety (PS). Furthermore, leader openness has been independently linked to PS (β = 0.32, *p* < 0.001) [46], reinforcing the importance of transparent and supportive leadership in fostering a psychologically safe work environment.

#### 4.4.2. Transformational Leadership

Transformational and charismatic leaders encourage individuals to prioritize collective goals over immediate self-interest by emphasizing communication and empowerment [47]. Researchers have found that transformational leadership positively influences transactive memory systems among healthcare workers, significantly predicting psychological safety (PS) (β = 0.70, t = 4.42, *p* < 0.001). PS predicts team learning behaviour [34]. Raes et al. [34] also investigated the relationship between laissez-faire leadership and PS. Unlike transformational leaders, who take a motivational and proactive approach, laissez-faire leaders often adopt a passive stance, avoiding using their authority to make decisions. Despite this passivity, laissez-faire leadership still demonstrated a positive association with TPS (β = 0.39, t = 2.34, *p* < 0.05), although its effect was weaker than that of transformational leadership. Both leadership styles foster PS by transferring agency to the group, allowing individual voices to be acknowledged and valued.

#### 4.4.3. Change-Oriented Leadership

Researchers have found that change-oriented leadership positively influences transactive memory systems at the organizational level [48]. Change-oriented leaders closely monitor industry trends and environmental shifts, identifying emerging opportunities and potential threats [37,48]. They actively encourage team members to challenge assumptions to drive continuous improvement, motivate employees to achieve ambitious goals, and take strategic risks to promote innovation and change [37]. In a cross-sectional study of healthcare workers, researchers analyzed how change-oriented leadership shapes healthcare teams’ learning processes and outcomes. Their findings revealed that while change-oriented leadership directly enhanced team learning behaviour and performance, psychological safety (PS) mediated this relationship. Specifically, they observed a significant positive association between change-oriented leadership and PS (β = 0.52, *p* < 0.001). The authors suggest that leaders who exhibit change-focused behaviours create a psychologically safe climate that facilitates team learning, potentially through team empowerment, cohesiveness, or collective efficacy. Furthermore, they propose that resilience plays a critical role in TPS leadership. They recommend that organizations integrate adaptability and change-management skills into selecting, promoting, training, and developing future team leaders and members [37].

### 4.5. Hierarchy and Structure

Hierarchical structures, while necessary for maintaining order and efficiency in complex systems, often create power differentials that can inhibit open communication and hinder the willingness of lower-ranking individuals to speak up, particularly when their input challenges authority figures [28]. In healthcare, for instance, rigid hierarchies have been linked to medical errors, as junior staff may hesitate to question senior physicians even when they identify potential safety concerns [49]. Similarly, in law enforcement, command-driven structures may discourage officers from reporting misconduct or voicing operational concerns, reinforcing a culture of silence that can compromise ethical decision-making and organizational accountability [50]. Research suggests that fostering a culture of psychological safety requires leaders to adopt inclusive and supportive leadership styles, actively encouraging input from all levels of the organization and reducing fear-based communication barriers [14,25,31]. When high-risk professionals feel psychologically safe, individuals are more likely to report safety hazards, challenge problematic norms, and contribute to continuous improvement efforts, ultimately enhancing individual and organizational performance [10,31]. Therefore, leaders must balance hierarchical structures with practises that promote trust, respect, and psychological safety to improve operational efficiency and crisis management while mitigating the risks of rigid power dynamics [1].

### 4.6. Hierarchy Impact on Leadership

Workplace hierarchy refers to the structured arrangement of individuals within an organization or team based on power, status, and job function [14]. This hierarchical structure can significantly influence transactive memory systems, as demonstrated in multiple studies [12,30]. In contrast to more participatory leadership styles, autocratic organizations maintain a rigid hierarchical structure characterized by centralized authority and authoritarian management, where decision-making power resides primarily with the leader [31,51]. While autocratic leadership can negatively impact employee performance and morale—potentially leading employees to feel undervalued or subordinate—specific contexts demonstrate its capacity to enhance TPS [30,52]. For instance, high-risk industries such as the military, public safety organizations, hospitals, and airlines typically maintain a well-defined hierarchical structure, often delineated by job titles or ranks influencing individuals’ perceived status within the organization [30]. Although higher-ranking individuals ultimately hold decision-making authority, subordinates experience enhanced TPS when they perceive that their superiors acknowledge their voices despite existing status differentials [12,30].

The broader organizational context in which teams operate plays a crucial role in shaping team dynamics and performance. Bresman and Zellmer-Bruhn [51] propose that the positive relationship between autocratic leadership and TPS may depend on whether the team perceives the imposed structure as legitimate and beneficial, allowing for clearly defined roles, responsibilities, and decision-making rules. Similarly, Diabes et al. [31] found that healthcare professionals with well-defined team roles demonstrated higher TPS, reinforcing that role clarity contributes to effective knowledge coordination and team functioning.

### 4.7. Role of Hierarchical Structures

Research on lower-risk occupations suggests that team-imposed structural organization fosters a shared understanding of operational norms, thereby reducing uncertainty and cultivating a TPS-supportive climate [51]. However, when organizations impose structure externally, constraints on task autonomy at the organizational level may diminish TPS [52].

In a study examining the rank-and-file personnel within a Canadian police organization, researchers found that perceptions of fair treatment, particularly in equitable justice, were positively associated with transactive memory systems. This association, in turn, enhanced employees’ engagement in their work [12]. Specifically, employees who experienced psychological safety demonstrated greater task focus, exhibited a higher commitment to their responsibilities, confidently shared ideas, and perceived greater meaning in their work. Notably, this effect was more pronounced among police officers than civilian employees, suggesting that law enforcement’s hierarchical and operational structures may influence the relationship between psychological safety and workplace engagement [12].

Building on these findings, the interplay between hierarchical structures and psychological safety in high-risk occupations suggests that while structure provides clarity and operational efficiency, excessive rigidity may inhibit the development of transactive memory systems and diminish employees’ willingness to engage in open communication. Hierarchies in professions such as law enforcement and healthcare often emphasize chain-of-command decision-making. This can create a culture where subordinates hesitate to voice concerns due to fear of repercussions or perceived futility [1]. In contrast, research suggests that participatory leadership and a climate of trust can mitigate these challenges by fostering psychological safety and enhancing employees’ ability to share expertise and coordinate effectively in complex, high-stakes environments [25]. Moreover, when organizational structures allow for collaborative decision-making, employees feel a greater sense of agency, strengthening their commitment to their roles and improving team cohesion [28]. Thus, the effectiveness of TPS in high-risk environments depends not only on the presence of hierarchical structures but also on the degree to which those structures facilitate rather than hinder open dialogue and mutual learning among team members.

Furthermore, the moderating role of professional identity in high-risk settings underscores the differential impact of organizational hierarchy on TPS. Research on healthcare teams, for instance, has demonstrated that psychological safety is critical in environments where interdisciplinary collaboration is required, as it enables professionals with varying levels of authority to contribute expertise without fear of professional repercussions [31,53]. Similarly, law enforcement officers who perceive their leadership as supportive are likelier to engage in knowledge-sharing behaviours, strengthening TPS and improving operational effectiveness [12,54]. However, rigid bureaucratic norms can impede this process by reinforcing top-down control mechanisms that limit autonomy and discourage feedback from lower-ranking personnel [50]. These findings highlight the importance of balancing structural discipline with inclusive leadership practises that promote psychological safety, ensuring that hierarchical systems do not undermine the cognitive and relational mechanisms essential for effective team functioning in high-risk occupations.

### 4.8. Empirically Tested Interventions to Enhance Psychological Safety

Although this review identified key antecedents of psychological safety (PS) in high-risk occupations, limited attention has been paid to empirically tested interventions to foster PS actively. The literature has increasingly explored practical strategies that organizations can implement to enhance PS, suggesting that interventions at the leadership, team, and organizational levels are critical. Structured training programmes have shown promise in strengthening psychological safety by enhancing inclusive and ethical leadership behaviours. For example, intervention studies, such as Kolbe et al. [31], demonstrated that debriefing techniques like TeamGAINS, which emphasize leader inclusiveness and structured team reflection, significantly improved perceptions of PS and teamwork effectiveness in healthcare simulation settings [31]. Similarly, Ridley et al. [14] found that structured teamwork training in a high-risk surgical environment significantly improved speaking-up behaviours and perceptions of teamwork climate, highlighting the efficacy of formalized communication and collaboration interventions [13].

At the team level, interventions such as post-event debriefings, feedback mechanisms, and error-reporting systems have been associated with improved psychological safety climates. Structured debriefings allow for reflection on team performance in a non-punitive environment, normalizing interpersonal risk-taking and reinforcing learning behaviours [3,31]. Research indicates that repeated exposure to psychologically safe debriefings fosters greater resilience and trust within high-risk teams [31].

Organizational strategies are equally critical. Policies that formalize whistleblower protections, promote just culture frameworks, and embed inclusivity into recruitment and advancement practises contribute to sustained improvements in PS [22]. Initiatives that establish explicit expectations for respectful communication, empower employees to voice concerns, and recognize contributions to team learning are particularly effective in mitigating the hierarchical barriers identified in this review. Despite these advances, the empirical evaluation of PS interventions remains underdeveloped, with many studies relying on pre-post designs lacking control groups, limiting causal interpretations. Future research should prioritize] randomized controlled trials (RCTs) and longitudinal studies to rigorously test the effectiveness of interventions in enhancing psychological safety over time across diverse high-risk settings.

### 4.9. Climate and Culture

Organizational climate and culture represent interrelated constructs that shape employees’ experiences and perceptions of their workplace social environment [9]. Organizational culture encompasses the shared beliefs, assumptions, and values embedded within an organization, shaping its norms and guiding behaviours [2]. In contrast, organizational climate reflects employees’ interpretations of policies, practises, procedures, and expectations regarding behaviour, including how behaviours are reinforced and rewarded [55]. These constructs collectively influence workplace dynamics, particularly in high-demand professions such as healthcare, where emotional resilience is essential for sustaining employee engagement and performance.

### 4.10. Caring Climate

A caring organizational climate, characterized by compassion and interpersonal support, plays a crucial role in mitigating workplace stress and burnout, particularly in professions that involve continuous caregiving [27]. Healthcare professionals who frequently encounter emotionally demanding situations experience higher rates of burnout and emotional exhaustion than individuals in other fields. As such, fostering a work environment that prioritizes care and support for patients and employees is essential in reducing these adverse effects. Empirical findings suggest that healthcare workers in a supportive and caring organizational climate report lower emotional exhaustion levels than those in environments lacking such support. Furthermore, employees with lower levels of job empowerment appear to derive the most significant benefit from a caring climate, reinforcing its importance in workplace well-being [27]. The mechanism underlying this positive effect is attributed to increased transactive memory systems, with research indicating a significant relationship between a caring climate and TPS (β = 0.53, *p* < 0.001), moderated by worker empowerment. These findings suggest that a healthcare work environment fostering mutual care and support among employees can enhance TPS, ultimately improving employee well-being and overall organizational effectiveness [27].

A caring organizational climate also significantly influences psychological safety, essential for fostering open communication, teamwork, and a sense of inclusion within healthcare settings. Research has demonstrated that when healthcare professionals perceive their workplace as supportive and compassionate, they are more likely to feel psychologically safe, encouraging proactive problem-solving, knowledge sharing, and innovation [9,25]. Conversely, in environments where care and interpersonal support are lacking, employees may hesitate to voice concerns or suggest improvements, fearing retaliation or professional repercussions [28]. Rathert et al. [27] assert that a strong, caring climate fosters employee trust, enhances job satisfaction, and mitigates the detrimental effects of hierarchical power structures that can stifle communication and collaboration. As a result, organizations prioritizing psychological safety through a caring climate improve employee well-being and patient outcomes by encouraging healthcare teams to communicate openly and effectively [27].

Moreover, the relationship between a caring climate and psychological safety extends beyond individual well-being to impact organizational resilience and performance [25]. Studies indicate that healthcare organizations with high psychological safety exhibit lower turnover rates, higher job engagement, and improved teamwork efficiency [9,26]. This is particularly relevant in high-stress environments, where collaborating effectively under pressure is vital for patient safety and quality of care. A caring climate reinforces psychological safety by validating employees’ contributions, promoting fair treatment, and ensuring that leadership actively supports staff well-being [1]. When employees trust that their colleagues and supervisors will help them in difficult situations, they are more likely to engage in proactive behaviours that strengthen team cohesion and enhance TPS [56]. Consequently, fostering a workplace culture that integrates psychological safety and a caring climate can lead to more resilient and adaptive healthcare teams, ultimately improving employee retention and patient care quality [26,27]. These findings emphasize embedding compassionate leadership and supportive workplace policies within healthcare institutions to sustain psychological safety and long-term organizational success [9,26,57].

### 4.11. Empowerment Through Communication

In alignment with previous scholarly findings, our research underscores communication as the central mechanism for establishing and maintaining transactive memory systems within organizations [56]. This process encompasses formal and informal communication channels, including peer-to-peer interactions, leadership discourse with employees, and the policies and regulations that structure organizational functioning. Effective communication fosters a shared cognitive framework, facilitates coordination, and enhances trust, all of which are integral to optimizing TPS. Comparative analyses of communication patterns in healthcare settings characterized by varying levels of psychological safety reveal significant disparities. Yanchus et al. [54] found that healthcare professionals operating in environments perceived as psychologically safe reported greater ease in expressing concerns without apprehension of punitive repercussions, engaging in objective and constructive professional discourse, and benefiting from frequent and transparent communication practises. Moreover, critical elements such as the presence of an involved and supportive supervisor, the use of respectful and positive verbal interactions, and access to private spaces for discussing sensitive patient information were identified as key contributors to psychological safety.

Additionally, the quality of interpersonal communication, characterized by openness, honesty, and mutual trust between employees and management, emerged as a fundamental determinant in strengthening TPS. When these communication dynamics were present, teams demonstrated greater cohesion, knowledge-sharing efficacy, and workplace engagement. Conversely, environments, where team members responded to disagreements with defensiveness or engaged in unconstructive, personalized conflict exhibited a decline in TPS effectiveness. This suggests that professionalism and objectivity in workplace communication are critical for sustaining TPS, particularly in high-stakes environments such as healthcare. Since TPS facilitates efficient knowledge coordination and retrieval, maintaining high-quality communication standards is essential for optimizing team performance and ensuring safe and effective patient care [56].

Psychological safety, reinforced through open and transparent communication, is a critical determinant of employee empowerment, particularly in high-risk professional settings such as healthcare [57]. When employees feel psychologically safe, they are more likely to voice their concerns, offer innovative solutions, and engage in collaborative problem-solving without fear of negative consequences [5,9]. Communication plays a pivotal role in shaping this environment, as leaders who foster a culture of inclusive dialogue and active listening empower employees by validating their contributions and reinforcing their sense of agency [25,56]. Research suggests that in psychologically safe workplaces, employees perceive their input as valued, which enhances their engagement and willingness to assume responsibility for team outcomes [1,40]. Furthermore, communication-driven empowerment increases job satisfaction and reduces burnout as employees gain greater autonomy in decision-making and develop a stronger sense of professional fulfilment [56]. This dynamic is particularly significant in healthcare environments, where hierarchical structures sometimes stifle lower-ranking employees from expressing their perspectives [12,31]. By promoting a culture in which transparent and respectful communication is the norm, organizations can foster psychological safety and, in turn, bolster employees’ confidence in their professional roles [9,31].

Moreover, effective communication enhances psychological safety and facilitates knowledge-sharing, skill development, and organizational learning, all contributing to employee empowerment [58]. Within transactive memory systems, the seamless exchange of information among team members ensures that expertise is distributed efficiently, reducing cognitive overload and fostering collective intelligence [59]. This process is contingent upon an organizational climate in which communication is encouraged, and employees feel safe to seek clarification, share knowledge, and challenge assumptions without fear of reprisal [28]. Empirical evidence indicates that when healthcare professionals operate in environments where communication is open and constructive, they are more likely to engage in proactive learning behaviours, which strengthens their competencies and overall job performance [13,29]. Conversely, in workplaces where communication barriers exist, employees may feel isolated, hesitant to ask for guidance, and disengage from their teams, undermining psychological safety and TPS [54]. Consequently, fostering an environment where prioritized communication enhances individual empowerment and strengthens organizational resilience and adaptability, ultimately improving patient care and overall operational efficiency [26].

## 5. Limitations

There are several limitations worth noting in this scoping review. This review did not include studies from collectivist countries, as the focus was on workplace psychological safety (PS) in predominantly individualistic societies. While this approach ensures consistency in examining psychological safety within similar organizational structures, it limits the applicability of the findings to cultures where teamwork, group harmony, and hierarchy may influence PS differently. Future research should explore how psychological safety manifests in collectivist contexts, as existing studies may offer valuable insights into different cultural dynamics.

Research on TPS in high-risk populations, such as emergency response teams and military personnel is lacking. While the findings contribute to understanding TPS in healthcare and public safety professions, they may not fully capture underrepresented high-risk occupations’ unique challenges. Additional research is needed to examine how psychological safety is influenced by factors such as combat environments, rapid decision-making under extreme stress, and strict hierarchical structures in these settings.

The search strategy had several limitations. While comprehensive and guided by established scoping review methodologies, it was restricted to studies that explicitly examined psychological safety. Broader literature on related topics, such as workplace stress, organizational justice, and employee well-being, was not included unless psychological safety was a primary focus. As a result, caution should be exercised in interpreting these findings, as other relevant evidence may exist that indirectly addresses psychological safety in high-risk workplaces.

Additionally, the sample of included studies was relatively small. This reflects the emerging nature of the literature specifically addressing psychological safety in high-risk sectors like healthcare and public safety. While research on related workplace constructs is extensive, empirical studies that measure psychological safety directly using validated instruments in these sectors remain limited, resulting in a smaller pool of eligible studies for inclusion.

A further limitation of this scoping review lies in the influence of contextual factors—such as culture and organizational policy—on the generalizability of findings. While the review focused on studies from predominantly individualistic societies to ensure conceptual consistency, this restricts applicability across diverse cultural contexts. Psychological safety (PS) may manifest differently in collectivist cultures, where group harmony and deference to authority often shape workplace behaviours [3]. Moreover, variations in organizational structures—such as speaking-up mechanisms and hierarchical communication norms—can influence the relationship between PS antecedents and outcomes [3,22]. Thus, while factors like inclusive leadership and supportive climates appear consistently linked to PS in high-risk occupations, these conclusions should be applied cautiously across differing cultural and organizational environments. Future research should examine how sociocultural and institutional contexts interact with PS to shape diverse workplace outcomes.

Another limitation is the reliance on published, peer-reviewed literature, which introduces the potential for publication bias. Grey literature, unpublished studies, and research available in non-English languages were excluded, meaning some relevant findings may have been overlooked. Additionally, variations in study methodologies, sample sizes, and outcome measures posed challenges in synthesizing results, as differences in research design may affect comparability.

### Gaps in the Literature

This review identified several critical gaps in the existing TPS literature. Future research should explore the intersection between leadership behaviours, organizational structure, and workplace culture to develop targeted interventions to foster psychological safety in high-risk work environments. Additionally, longitudinal studies could provide more insight into the causal relationships between TPS and workplace outcomes, offering a more comprehensive understanding of how TPS can be sustained over time.

Longitudinal Studies: Most studies were cross-sectional, limiting the ability to assess causality. Longitudinal research is needed to understand the long-term effects of TPS antecedents.Diversity Considerations: Few studies explored how gender, race, or cultural backgrounds influence TPS. Future research should examine the intersectionality of diversity and TPS outcomes.Underrepresented Sectors: While healthcare and law enforcement were well-studied, other high-risk professions, such as military, firefighting, and emergency response teams, were underrepresented. Expanding TPS research to these sectors is necessary.Intervention Studies: Limited experimental research exists on interventions aimed at enhancing TPS. Future studies should evaluate specific organizational programmes to improve psychological safety in high-risk environments.

## 6. Conclusions

This scoping review has synthesized the existing literature on team psychological safety (TPS) in high-risk occupations, particularly within the healthcare and public safety sectors. The findings illustrate that TPS is crucial to workplace outcomes, influencing employee well-being, performance, error reporting, and overall team effectiveness [5,56]. Leadership emerged as the most significant antecedent of TPS, with inclusive, transformational, ethical, and change-oriented leadership styles fostering environments where employees feel safe to communicate openly, share concerns, and contribute to team learning [13,23,25]. Organizational structures and hierarchies play a complex role in shaping TPS, with rigid, command-driven structures sometimes acting as barriers to open dialogue and risk-taking [12,31]. However, psychological safety can be reinforced when leadership balances hierarchical control with participatory decision-making, ensuring employees feel included in decision-making processes without compromising necessary structural order [10,37]. Furthermore, workplace climate and organizational culture significantly affect TPS, with caring, just, and transparent work environments positively associated with higher levels of psychological safety [1,26]. The review highlights that fostering TPS in high-risk environments requires a multifaceted approach. Leadership behaviours prioritizing trust, fairness, and inclusivity are essential in establishing an atmosphere where employees feel valued and empowered [3,56]. Additionally, reducing authoritarian tendencies in hierarchical structures while maintaining clear role expectations and accountability mechanisms can enhance TPS and facilitate more effective team collaboration [12,30]. The evidence suggests that when employees perceive their workplace as psychologically safe, they are more likely to report errors, seek feedback, and contribute to innovation, ultimately improving organizational outcomes [7,10,34]. 

Despite these insights, several gaps in the literature remain. First, while many studies establish correlations between leadership styles and TPS, fewer have explored causality through longitudinal designs [13,25]. Future research should employ experimental and longitudinal methodologies to better understand the mechanisms through which TPS is developed and sustained over time [31,56]. Additionally, diversity considerations have been underexplored, with few studies investigating how gender, race, or cultural backgrounds shape perceptions of psychological safety [60]. Future research should examine how inclusive leadership practises and organizational policies can actively foster psychological safety for underrepresented groups, ensuring that all employees feel valued, heard, and empowered to contribute [61]. Addressing these gaps is critical, given the increasingly diverse nature of the global workforce. Moreover, while healthcare and law enforcement have been extensively studied, other high-risk professions, such as firefighting, emergency response, and the military, warrant further investigation to determine how TPS functions in these contexts [14].

Interprofessional teams present another area for future research. While the review identifies leadership as crucial in promoting TPS, interprofessional teams often consist of multiple leaders fulfilling distinct roles [33]. Understanding how different leadership styles interact within interprofessional teams could provide further insights into optimizing team performance [34]. Additionally, research should explore how leadership training programmes can be tailored to different organizational contexts to enhance TPS more effectively [25,37].

The findings of this review have several practical implications. Organizations should prioritize leadership development programmes emphasizing inclusivity, ethical integrity, and transformational leadership behaviours [3]. Moreover, structural changes that promote fair decision-making, transparency, and a culture of open dialogue can significantly improve TPS [12]. Implementing structured feedback mechanisms, fostering mentorship relationships, and promoting a culture that normalizes learning from mistakes rather than punishing them are additional strategies that can enhance psychological safety in the workplace [9,31].

Overall, this scoping review underscores the importance of TPS as a foundational element of effective team dynamics in high-risk occupations. It provides a comprehensive overview of the factors influencing TPS and offers a roadmap for future research and organizational practice. By investing in leadership development, fostering supportive workplace climates, and balancing hierarchical structures with inclusive decision-making, organizations can cultivate environments where employees feel psychologically safe, ultimately leading to improved performance, well-being, and organizational resilience [56].

## Figures and Tables

**Figure 1 ijerph-22-00820-f001:**
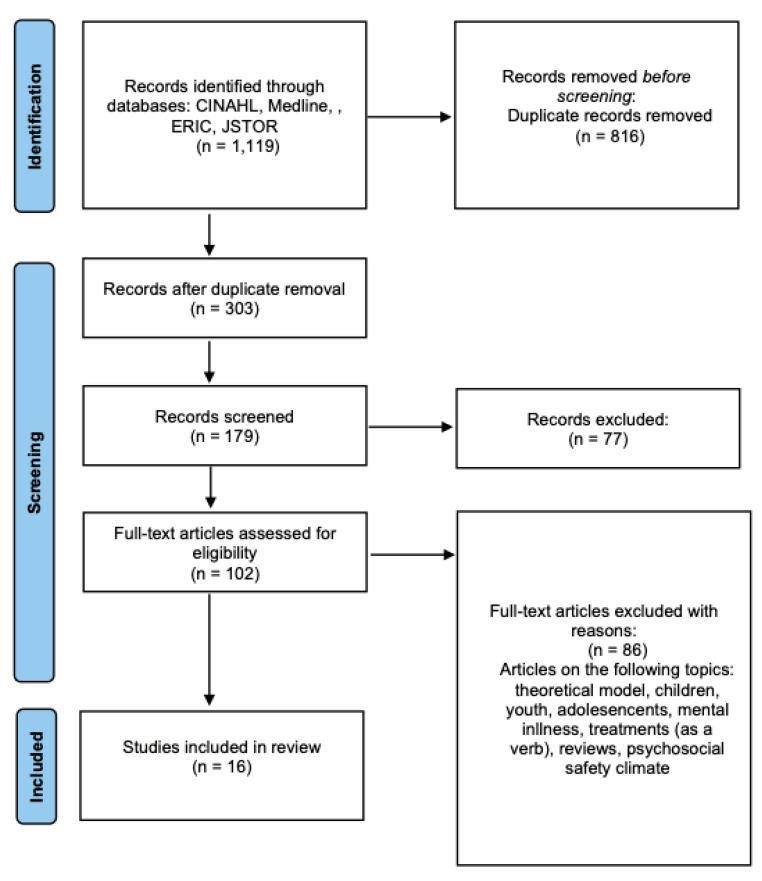
PRISMA diagram [21].

**Table 1 ijerph-22-00820-t001:** SPIDER search strategy.

Spider Tool	Search Terms
Sample	“public safety worker” OR “firefighter” OR “police officer” OR “nurse” OR “Physician” OR “occupational therapist” OR “health and social care” OR “supervisor” OR “leader” OR “adult” OR “care worker” OR “support worker” OR “organization” OR “employee”
Phenomena of Interest	“psychological safety” or “interpersonal risk” or “psychological safe” OR “team” OR “psychological safety climate” OR “job” or “workplace safety” or “workplace culture” OR “intervention” OR “teamwork”
Design	“systematic review” OR “literature review” OR “scoping review” OR “longitudinal” OR “thematic analysis” OR “ethnography” OR “survey”
Evaluation	“outcome” OR “intervention” OR “job satisfaction” OR “safety” OR “experience”
Research Type	“quantitative” OR “qualitative” OR “mixed-methods”

**Table 2 ijerph-22-00820-t002:** Inclusion and Exclusion Criterion.

Inclusion Criteria	Exclusion Criteria
Articles published from 2010–2024	Studies set in collectivist countries
Studies published in peer-reviewed journals and outlining primary empirical studies	Populations that included children; and the settings of low-risk workplace environments
Grey literature such as newspaper articles may be included if supporting peer review findings	Successive studies to TPS
Examined antecedent studies to TPS	Studies not published in the English language
Measured TPS using a validated questionnaire	Non-measured TPS results
Population studied included adult workers; and the setting was a high-risk workplace environment	
Studies published in the English language

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
