# Peer review of "Antecedents of Workplace Psychological Safety in Public Safety and Frontline Healthcare: A Scoping Review"

_ijerph, 2025, doi:10.3390/ijerph22060820_

Round 1

Reviewer 1 Report

Comments and Suggestions for Authors

The article thoroughly synthesizes a diverse body of literature, offering a clear conceptual framework for understanding psychological safety antecedents in high-risk settings. 

However, authors should consider the following:

  • The review includes only 16 studies, which, while carefully selected, may limit the comprehensiveness of conclusions. This relatively small sample size reflects the nascent state of research but also suggests that findings should be interpreted with caution.

  • The article relies heavily on qualitative thematic analysis, which, although appropriate for a scoping review, limits the ability to quantify the strength of relationships between antecedents and psychological safety outcomes.

  • There is limited discussion on how contextual factors (e.g., cultural differences between countries, specific organizational policies) might influence the generalizability of findings.

  • The review could benefit from a more detailed exploration of interventions and practical strategies that have been empirically tested to enhance psychological safety.

Author Response

Please see the attachment. As the system would only allow one attachment, we combined the feedback and updated manuscript into one document. Thank you!

Reviewer 2 Report

Comments and Suggestions for Authors

Dear Authors,

Thank you for the opportunity to read your work and contribute, through my comments and suggestions, to its possible improvement. My first words go to the quality of your research and the detailed and comprehensive way in which you present your results. In my view, this aspect should be emphasised.

Below, I detail a few points I encourage the authors to develop:

1. On page 3, line 89, the authors state, "Research has shown that fostering TPS in these environments can enhance learning, adaptability...". Perhaps it would be better to identify which research this is.

2. On page 6, section 2.5. Inclusion and Exclusion Criteria, in my opinion, your argument for excluding articles from ‘collectivist countries’ should be developed. I know it's more than common to categorise most European countries, the United States, or Canada as individualistic societies, right, it's a way of seeing it like any other. But does saying that this is an inclusion criterion mean that, beforehand, we know that culture influences the level of psychological safety in the workplace? Am I right? If so, in my opinion, the authors should identify the research that corroborates this position. On the other hand, although in my view this inclusion/exclusion criterion is debatable, doesn't its assumption mean that psychological safety is determined by culture more than by any other factor? If this is the case, intervention at the individual and organisational levels would be pointless, which would actually greatly limit the possibilities for concrete change. 
I would like to stress that my comment is not intended to contest the authors' choice, but only to instigate critical reflection.

3. As for the studies reviewed, it's a little difficult to understand why you included a study in the aviation sector (mentioned on page 8 and then on page 21). Looking at the title of the manuscript, its objective and above all the search strategy (on page 5), it's not clear why you included studies with aircrew members. Since these workers are not considered "public safety" or "frontline healthcare", what is the reason for including them in the research?

4. I strongly suggest that the authors rework the table summarising the results (the table starting on page 11), especially the column dedicated to the results of each study. Given the length of this table, and above all the variability with which the results of each study are summarised, it is difficult for the reader to follow this results presentation. In my opinion, the authors should reduce this table and define a common strategy for summarising the results of each article reviewed.

5. Once again, the exhaustive way in which the authors discuss the results should be noted (in section 4. Discussion). However, in my view, the authors should revise this section and perhaps create a subsection dedicated to practical implications for organisational decision-makers. I mean 4 or 5 takeaways, or strong messages, that could guide organisational decision-makers who want to intervene in this matter. This is not to say that the authors don't mention these practical implications, because they are there; but my suggestion is to give them greater prominence in the article.

6. Finally, the subsections' numbering of the Discussion section should be revised, since there are minor inaccuracies.

Congratulations and all the best. 

Author Response

(The authors gave the same response as above.)
